# Development and evaluation of three automated media pooling and molecular diagnostic systems for the detection of SARS-CoV-2

Yasufumi Matsumura,[1] Taro Noguchi,[1] Koh Shinohara,[1] Masaki Yamamoto,[1] Miki Nagao[1]

**ABSTRACT**  Pooled testing combined with molecular diagnostics for the detection of SARS-CoV-2 is a promising method that can increase testing capacities and save costs. However, pooled testing is also associated with the risks of decreased test sensitivity and specificity. To perform reliable pooled testing, we developed and validated three automated media pooling and molecular diagnostic systems. These pooling systems (geneLEAD-PS, Panther-PS, and Biomek-PS) comprised existing automated molecular detection platforms, corresponding automated media pooling devices, and laboratory information management systems. Analytical sensitivity analysis and mock sample evaluation were performed, and the obtained data were used to determine the sizes of the pool for the validation study. In the validation study, a total of 2,448, 3,228, and 6,420 upper respiratory samples were used for geneLEAD-PS, Panther-PS, and Biomek-PS, respectively, and the diagnostic performances were compared with the reference RT−PCR assay. A pool size of 6 for geneLEAD-PS and a pool size of 4 for Panther-PS and Biomek-PS were selected for the validation studies. All three systems showed high positive percent agreement values of ≥90.5% and negative percent agreement values of ≥99.8% for any specimen type. Pooled testing resulted in a 65%–71% reduction in cost per sample. The testing capacities of geneLEAD-PS, Panther-PS, and Biomek-PS were 144 samples in 3 hours, 384 samples in 5.5 hours, and 376 samples in 4 hours, respectively. The developed pooling systems showed robust diagnostic performances and will increase the testing capacities of molecular diagnostic tests while saving costs and may contribute to infection control of COVID-19.

**IMPORTANCE**  During the COVID-19 pandemic, there have been surges in demand for accurate molecular diagnostic testing and laboratory supply shortages. Pooled testing combined with highly sensitive molecular testing, which entails mixing multiple samples as a single sample, is a promising approach to increase testing capacities while reducing the use of consumables. However, pooled testing is associated with risks that compromise diagnostic performance, such as false negatives due to dilution of positive samples or false positives due to cross-contamination. To perform reliable pooled testing, three different pooling systems (an automated pooling device, an automated molecular detection platform, and a laboratory information management system) were developed to accurately interpret pooled testing results. These three systems were validated using multiple clinical samples and showed high concordance with individual testing. The developed pooling systems will contribute to increasing reliable molecular testing capacities while using fewer consumables and saving costs.

**KEYWORDS**  COVID-19, SARS-CoV-2, pooling, automation, liquid handling instrument

**Ad Hoc Peer Reviewer**  Xianding Deng

Address correspondence to Yasufumi Matsumura, yazblood@kuhp.kyoto-u.ac.jp.

Y.M. received research funds from Beckman Coulter, Nippon Control System, Precision System Science, and Toyobo. M.N. received research funds from Beckman Coulter and Precision System Science.

See the funding table on p. 12.

At least one-third of SARS-CoV-2 infections are asymptomatic (1). SARS-CoV-2 can be transmitted not only by individuals with coronavirus disease (COVID-19) but also by asymptomatic carriers (2). The establishment of rapid, accurate, and cost-effective large-scale screening and surveillance strategies for certain populations, including individuals at high risk of infection (e.g., essential workers, cross-border returnees), inpatients who are at risk of nosocomial infection, and individuals in infection clusters, is important for the identification of asymptomatic carriers.

Pooled testing methods, in which multiple samples are pooled together as a single sample, theoretically increase the number of tests that can be performed simultaneously. For example, at a prevalence of 5%, a pool size of 5 would result in the greatest reduction (57%) in the expected number of tests (3). The combination of pooled testing with highly sensitive molecular testing can lead to reductions in testing reagents, costs, and labor (4). Strategies to save reagents are also important under prolonged global supply shortages, which have occurred during the COVID-19 pandemic (5). Previous studies have shown that the results of pooled testing were consistent with those of individual testing and that pooled testing leads to a reduction in test sensitivity within theoretical calculations (4, 6). However, the majority of these studies involved only a mock sample evaluation. Large-scale pooled testing (approximately 40,000–250,000 samples) has already been implemented in clinical laboratories or screening programs for schools or universities using pool sizes of 5, 8, 10, or 24, depending on expected test positivity (7–9). China utilized pooled testing with pool sizes of 3, 5, or 10 for community screening that targeted >1 million residents in moderate-risk areas, while people in high-risk areas were tested individually to control local outbreaks (10, 11). However, in these large-scale reports, the diagnostic performance of these methods was not investigated due to a lack of comparisons with individual tests.

There are some concerns with the pooling method, including decreased sensitivity due to negative samples diluting positive in the pool, the potential risk of decreased specificity due to cross-contamination, and longer turnaround times for samples in a pool that yields positive results and thus requires retesting. Pooling efficiency depends on the pool size and prevalence of infections (12). A higher pool size and lower prevalence result in higher efficiency but the maximum size of the pool needs to be limited depending on the molecular assays used to achieve an acceptable level of clinical sensitivity (13). A survey of the US laboratories showed that the barriers to pooled testing included lack of clear protocols, adequate resources for testing and validation, and difficulty in performing pooled testing workflow (e.g., retesting process of individual samples and a lack of the necessary software to track samples through the workflow) (14).

Laboratory automation is another important approach to scale up testing capacity (13). It can contribute to reducing errors and improving quality while saving labor and costs (15). Automation can also contribute to accurate testing of pooling procedures by providing accurate mixing, sample tracking, and avoidance of cross-contamination. In previous reports, the pooling and testing stages were usually performed manually (or methods not specified), although some studies utilized automated extraction devices or fully automated platforms for the testing of pooled samples (4, 16–19). Barak et al. utilized a liquid handler for automated media pooling but its performance was not validated by comparison with individual testing (7).

In this study, we developed three automated media pooling and molecular diagnostic systems for the detection of SARS-CoV-2, and their effectiveness was validated using clinical samples.

## RESULTS

The three developed pooling molecular diagnostic systems comprised a media pooling device, a fully automated molecular detection platform, and a laboratory information management system (LIS) (Fig. 1; Table 1): (1) geneLEAD pooling system (geneLEAD-PS); (2) Panther pooling system (Panther-PS); and (3) Biomek pooling system (Biomek-PS).

|  | **geneLEAD-PS** | **Panther-PS** | **Biomek-PS** |
|---|---|---|---|
| **Pooling stage**<br>Pooling device | 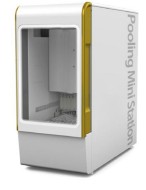<br>Pooling Mini Station<br>(Precision System Science) | 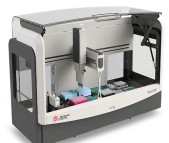<br>Biomek 4000<br>(Beckman Coulter) | 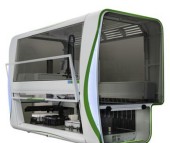<br>JANUS G3<br>(Revvity) |
| **Testing stage**<br>Fully-automated<br>molecular<br>detection<br>platform | 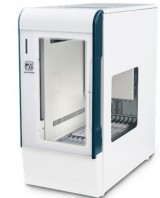<br>geneLEAD VIII<br>(Precision System Science) | 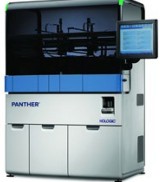<br>Panther<br>(Hologic) | 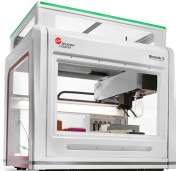<br>Biomek i5<br>(Beckman Coulter) |
| **Interpretation**<br>LIS | PSS LIS<br>(Precision System Science) | SimpPCR<br>(Nippon Control System) | SimpPCR<br>(Nippon Control System) |

**FIG 1** Three pooling systems were developed in this study. The geneLEAD pooling system (geneLEAD-PS) comprised Pooling Mini Station (Precision System Science, Chiba, Japan), geneLEAD VIII (Precision System Science), and PSS LIS (Precision System Science). The Panther pooling system (Panther-PS) comprised Biomek 4000 (Beckman Coulter, Tokyo, Japan), Panther system (Hologic Japan, Tokyo, Japan), and SimpPCR (Nippon Control System, Yokohama, Japan). The Biomek pooling system (Biomek-PS) comprised JANUS G3 Primary Sample Reformatter (Revvity Japan, Yokohama, Japan), Biomek i5 (Beckman Coulter), and SimpPCR (Nippon Control System). Each pooling device included a liquid handler and a barcode reader. LIS, laboratory information management system.

## Analytical sensitivity

For individual testing, geneLEAD-PS had the lowest limit of detection (LOD), followed by Panther-PS (nasal swab) and Biomek-PS (Table 2). The LODs of pooled testing increased depending on the sizes of pools of 4, 6, 8, or 10. The data used to calculate the LODs are shown in Table S1.

## Mock sample evaluation

At least 10 positive pools and 10 negative pools were tested for saliva and nasopharyngeal swab samples (Table 3). All systems produced correct individual and pooled testing results, except one saliva sample that was used for a pool size of 10, which was judged negative by Panther-PS (Data set S1).

## Air sampling

During the pooling procedures of each pooling device, five air samples were serially obtained. All the samples were negative according to the reference assay.

## Validation study

A total of 2,448 saliva, nasal swabs, or nasopharyngeal swabs, 3,228 nasal swabs, and 6,420 saliva, nasal swabs, or nasopharyngeal swabs were subjected to pooled testing using geneLEAD-PS, Panther-PS, and Biomek-PS, respectively. The pool sizes of 6 for geneLEAD-PS, 4 for Panther-PS, and 4 for Biomek-PS were selected. All three systems produced high positive percent agreement (PPA) values of ≥95%, negative percent agreement (NPA) values of ≥99.9%, and Kappa values of ≥0.95 (Table 4). Invalid results

**TABLE 1** Summary of the three developed pooling systems[i]

| Characteristic | geneLEAD-PS | Panther-PS | Biomek-PS |
|---|---|---|---|
| Specimen type | Nasopharyngeal swab, nasal swab, saliva | Proprietary nasal swab[a] (Aptima tube) | Nasopharyngeal swab, nasal swab, saliva |
| Pooling device | Pooling Mini Station[b] (PSS) | Biomek 4000[c] (BC) | JANUS G3 Primary Sample Reformatter[c] (Revvity) |
| Sample format | 2 mL tube on rack | Aptima tube on rack | Sampling tube on rack |
| Sample inactivation | Addition of 1/3 vol of lysis buffer, vortex, and incubate for 5 minutes | Vortex mix of the proprietary sample tube for 10 seconds | Addition of a mixture of 150 µL lysis buffer and 10 µL proteinase K solution[d] |
| Sample identification | Hand scanning of barcodes or barcode reading using PSS Barcode Reader | Hand scanning of barcodes | Automatic scanning of barcodes |
| Pool size used in the validation study (range) | 6 (2–8) | 4 (2–32) | 4 (2–10) |
| Pooling time for a maximum number of samples per batch[e] | 10 minutes for 48 samples | 20 minutes for 48 samples | 45 minutes for 192 samples |
| Automated molecular detection platform | geneLEAD VIII (PSS) | Panther (Hologic) | Biomek i5 (BC) with LightCycler 480 System II (Roche) |
| Sample input format | 2 mL tube on rack | Aptima tube on rack | 96-well deep-well plate |
| Sample input volume tested (volume for testing) | 400 µL (400 µL) | 2 mL (400 µL) | 200 µL (200 µL) |
| Sample identification | Hand scanning of barcodes | Hand scanning of barcodes | Automatic scanning of plate barcode |
| Nucleic acid extraction | MagDEA Dx SV (PSS) | Proprietary (Hologic) | RNAdvance Viral (BC) |
| Assay name | LeaDEA VIASURE SARS-CoV-2 (PSS) | Aptima SARS-CoV-2 (Hologic) | NIID N2 |
| Regulatory status | IVD | IVD | RUO-C |
| Testing time for 1 batch with a maximum number of samples (pools)[f] | 122 minutes for 8 samples | 330 minutes for 120 samples[g] | 141 minutes for 96 samples |
| Laboratory information management system | PSS laboratory information system[b] (PSS) | SimpPCR[b] (NCS) | SimpPCR[b] (NCS) |
| Pooling system configuration example (size of pool) | 1 Pooling Mini Station and 3 geneLEAD VIII (6) | 1 Biomek 4000 and 1 Panther (4) | 1 JANUS and 1 Biomek i5 (4) |
| Maximum number of samples and turnaround time | 144 samples, 3 hours for the first batch, 2 hours for the following batches | 384 samples, 5.5 hours; 1,100 samples, 8.5 hours | 376 samples, 4 hours for the first batch, 2.5 hours for the following batches |
| Cost per sample[h] (reduction rate) | $11 (76%) | $14 (69%) | $5 (69%) |

[a]Aptima Multitest Swab Specimen Collection Kit (Hologic), a sample tube filled with inactivation/preservation buffer and nasal swab for the Panther system, is recommended, but other specimen types, such as nasopharyngeal swabs or nasal swabs in viral transport media or saliva, can also be used when 1 mL of a sample is transferred to the Aptima sample tube containing 2.9 mL buffer (approximately fourfold dilution).

[b]The device and/or software itself was developed in this study through collaboration with the manufacturer.

[c]Specific customized protocols were developed in this study.

[d]JANUS consecutively performed the inactivation process following the pooling stage within a negative pressure booth.

[e]Pooling time was basically dependent on the number of samples, not the number of pools. The pooling time for the maximum number of samples at the size of the pool used in the validation study per batch was measured.

[f]The number of samples here implies that for individual testing. For pooled testing, the actual number of samples tested is calculated by the number of samples multiplied by the size of the pool selected.

[g]Panther allows random access to sample racks with a maximum of 15 samples and can load a maximum of 8 racks (120 samples). The result of the first sample is obtained after 210 minutes, and then the result of the next sample is obtained every minute.

[h]Average cost per 1 sample and cost reduction rate were calculated at an exchange rate of 120 yen = $1, at a prevalence (individual test positive rate) of 1%. Costs for reagents and consumables in both the pooling and individual testing stages were included.

[i]PSS, Precision System Science; BC, Beckman Coulter; NIID, National Institute of Infectious Disease in Japan; IVD, in vitro diagnostics; RUO-C, research use only but approved for clinical diagnostic use; NCS, Nippon Control System.

**TABLE 2** Analytical sensitivity of the three pooling systems

| Pooling system | Specimen type | Limit of detection[a], genome copies/mL sample (95% CI) | | | | |
|---|---|---|---|---|---|---|
| | | Individual testing | Size of pool | | | |
| | | | 4 | 6 | 8 | 10 |
| geneLEAD-PS | Saliva | 94 (81–131) | 524 (341–1,934) | 857 (553–2,676) | 908 (641–2,035) | ND |
| Panther-PS | Saliva | 1,323 (680–31,503) | 2,575 (1,487–18,200) | 5,552 (3,316–22,986) | ND | 16,124 (8,273–102,341) |
| Panther-PS | Nasal swab (Aptima) | 541 (337–2,024) | 1,778 (695–396,489) | 2,742 (1,534–9,099) | ND | 2,818 (1,636–8,912) |
| Biomek-PS | Saliva | 437 (272–1,602) | 2,328 (1,694–4,719) | 2,910 (2,099–7,851) | ND | 6,066 (2,982–21,301) |

[a]Calculated using a probit analysis. The data used for calculation are available in Table S1.
[b]ND, not determined; CI, confidence interval.

were not obtained in pooled testing (408, 807, and 1,605 pools) but were obtained in subsequent individual testing: 0.6% (2/324), 2.6% (4/156), and 0% (0/380) for geneLEAD-PS, Panther-PS, and Biomek-PS, respectively (Table S2). Retesting for these samples with invalid results could produce valid results. No significant differences in the PPAs and NPAs were observed among different sample types (Table 4). Ct value comparisons between pooled and individual samples are shown in Fig. S1.

## Cost and testing capacity

Theoretically, at a disease prevalence (test positive rate) of 1%, pooled testing could reduce the cost by 76% (geneLEAD-PS; size of pool of 6) or 69% (Panther-PS and Biomek-PS; size of 4) compared with individual testing (Table 1). In the validation study, at a prevalence of 1.0%–3.8%, depending on the specimens and systems used, pooled testing led to a 65%–71% reduction in the cost per sample (Table 4).

Testing capacities could be increased according to the size of the pool. Table 1 shows the pooling system configuration examples with the pool sizes used in the validation study and the turnaround times for the first and following batches. A longer turnaround time is needed when the first pool is large but the turnaround times for the subsequent batches were similar to that of individual testing because the pooling stage could be performed during the automated testing of the prior batch. For samples included in the positive pools, additional time was needed to test individual samples (Table 1). In the validation study, 85.4%–96.1% of samples were determined to be negative by only testing pooled samples (negative pool fraction; Table S2).

**TABLE 3** Diagnostic performance of the individual and pooled testing of the three pooling systems for mock samples[b]

| Specimen, size of pools | Number of positive/negative pools[a] | PPA/NPA | | |
|---|---|---|---|---|
| | | geneLEAD-PS | Panther-PS | Biomek-PS |
| Saliva | | | | |
| Individual testing | 10/190[a] | 100%/100% | 100%/100% | 100%/100% |
| 4 | 10/34 | 100%/100% | 100%/100% | 100%/100% |
| 6 | 10/14 | 100%/100% | 100%/100% | 100%/100% |
| 8 | 10/10 | 100%/100% | ND | ND |
| 10 | 10/10 | ND | 90%/100% | 100%/100% |
| Nasopharyngeal swab | | | | |
| Individual testing | 10/190[a] | 100%/100% | 100%/100% | 100%/100% |
| 4 | 10/29 | 100%/100% | 100%/100% | 100%/100% |
| 6 | 10/14 | 100%/100% | 100%/100% | 100%/100% |
| 8 | 10/10 | 100%/100% | ND | ND |
| 10 | 10/10 | ND | 100%/100% | 100%/100% |

[a]Number of samples is shown for individual testing.
[b]PPA, positive percent agreement; NPA, negative percent agreement; ND, not determined.

**TABLE 4** Diagnostic performance and costs of the three pooling systems in the validation study[e]

| Pooling system (size of pool) | Specimen | Number of positive/negative samples tested | Positive percent agreement, % (95% CI) | Negative percent agreement, % (95% CI) | Kappa (95% CI) | Cost per sample (reduction rate) |
|---|---|---|---|---|---|---|
| geneLEAD-PS (6) | All | 69/2,379 | 97.1 (93.1–99.7) | 99.9 (99.9–100) | 0.98 (0.95–1) | $14 (70%) |
| | Saliva | 31/1,403 | 100 (88.8–100) | 100 (99.7–100) | 1[a] | $16 (67%) |
| | Nasopharyngeal swab | 21/527 | 90.5 (77.9–100) | 100 (99.3–100) | 0.95 (0.87–1) | $14 (71%) |
| | Nasal swab | 17/449 | 100 (80.5–100) | 99.8[b] (99.3–100) | 0.97 (0.91–1) | $14 (69%) |
| Panther-PS (4) | Nasal swab (Aptima) | 45/3,183 | 95.6 (89.5–100) | 99.9[c] (99.8–100) | 0.95 (0.91–0.99) | $14 (69%) |
| Biomek-PS (4) | All | 106/6,314 | 98.1 (95.5–100) | 100 (99.9–100) | 0.99 (0.97–1) | $5 (69%) |
| | Saliva | 33/3,383 | 100 (89.4–100) | 100 (99.9–100) | 1[a] | $5 (71%) |
| | Nasopharyngeal swab | 26/1,271 | 100 (86.8–100) | 100 (99.7–100) | 1[a] | $6 (68%) |
| | Nasal swab | 47/1,660 | 95.7 (85.5–99.5) | 100[d] (99.8–100) | 0.98 (0.94–1) | $6 (65%) |

[a]95% CI could not be calculated.
[b]One pool that comprised negative samples tested positive, and one sample tested positive in subsequent individual testing.
[c]Two pools that comprised negative samples tested positive; in one pool, one sample tested positive in subsequent individual testing. All samples in the other pool were considered negative because they were negative in subsequent individual testing.
[d]Three pools that comprised negative samples tested positive, but these samples were considered negative because all samples were negative in subsequent individual testing.
[e]CI, confidence interval.

## DISCUSSION

### System development

The concept of pooled testing is based on mixing samples and making inferences about individual test results based on mixed samples (6). Pooled testing adds operational challenges and complicates the testing process. Borillo et al. claimed the importance of automated liquid handling for pooling and support from a LIS when implementing pooled testing in the laboratory (13). We developed pooled testing systems that included an automated liquid handler for the pooling stage and LIS to enable accurate handling of samples, sample tracking, and result interpretation and integration of pooling and testing stages (Fig. 1). These systems were developed in close cooperation with manufacturers, and this may support other laboratories in introducing pooled testing easily. In the testing stage, all systems used previously validated, automated molecular testing platforms for the testing stage to provide reliable molecular tests and to minimize labor. It is noted that minimum customization would enable configuration for pathogens other than SARS-CoV-2 by changing assays on the platform or the platform itself.

### Pooling procedures

We employed media pooling and a two-stage hierarchical method for pooling (20) as recommended by the U.S. Food and Drug Administration (FDA; https://www.fda.gov/medical-devices/coronavirus-covid-19-and-medical-devices/pooled-sample-testing-and-screening-testing-covid-19). This is the simplest method that has been most utilized in clinical settings (6). One-stage pooling procedures (such as matrix or array testing) (6, 20) were not adopted because they are vulnerable to cross-contamination, have difficulty recovering when the system crashes due to their complexity, and may be unable to identify positive samples from discordant testing results of pools that result in individual testing (6, 20).

### Analytical sensitivity

We previously investigated the analytical sensitivities of 11 commercial or in-house RT−PCR assays (based on nonautomated, manual methodology) with the same ATCC heat-inactivated strain used in this study and found that six assays achieved LODs of <500 copies/mL, followed by two assays with LODs of <1,000 copies/mL (21). Among four commercial direct RT−PCR assays that bypass the nucleic acid purification step, two assays showed LODs of 1,000–2,500, and the others showed LODs of >10,000 copies/mL (22). Based on these findings, we predefined the minimum LOD levels for individual and pooled testing of the developed automated systems as 1,000 and 2,500 copies/mL, respectively, to achieve similar levels of performance to manual RT-PCR assays or direct RT-PCR assays. All three developed systems are compatible with respiratory specimens, including saliva and swabs, but saliva was chosen as the primary matrix due to its relative difficulty in processing (usually more viscous and/or larger amount of human-derived nucleic acids). The LOD of Panther-PS was the highest among the three systems and was above 1,000 copies/mL due to its fourfold dilution of samples in the proprietary testing tube. Therefore, we determined that saliva was not suitable for Panther, and we added the analytical sensitivity analysis for the proprietary nasal swab, the default specimen of the Panther system that does not require specimen dilution (Table 1). This nasal swab test by Panther-PS showed a LOD <1,000 copies/mL. The LODs of all three systems for pooled testing were within the expected ranges (a LOD for individual testing multiplied by the size of the pool was included within the 95% CI of the pooled testing). These results indicate that appropriate pooling procedures were performed.

### Size of pool

Various sizes of pools (3–48) have been studied (4, 6). The optimal size of the pool depends on the settings including the prevalence and the sensitivity of the pooled

testing system. Daniel et al. proposed the use of pooled testing with a pool size of ≤5 when the expected prevalence was <5% but a pool size of ≤10 might be used when the expected prevalence was <1% according to data from validation studies (6). The developed systems can accept various sizes of pools (Table 1). To conduct the validation study, we chose the sizes of the pool according to the analytical sensitivity, the results from mock sample evaluation, and the estimated negative pool fraction of >90%. The last parameter was set because we considered that turnaround time was one of the important factors for screening testing for COVID-19. Early identification of disease-negative persons would result in stopping unnecessary infection control practices that were performed while waiting for testing results. In addition, an increase in the number of individual tests for positive pools would decrease the testing capacity for the next pooled samples, resulting in longer turnaround times. In accordance with the above criteria, a pool size of 6 was selected for geneLEAD-PS. In the validation study, the prevalence (2.2%–3.8% according to sample types; Table S2) was higher than the preestimation of 1.4%, which resulted in a lower negative pool fraction (85.4%–88.8%) than 90%. Panther-PS and Biomek-PS used a pool size of 4; the prevalence in the validation study was <2.8%, and the negative pool fraction was >90%. Cost-saving effects were similar (approximately 70%) between geneLEAD-PS and the others. Based on these results, we suggest using the pool sizes used in the validation study if a prevalence of approximately <3% is expected and if >85% of the negative pool fraction is acceptable.

## Validation study

Previous validation studies that employed different methods (e.g., size of the pool, assays) reported variable diagnostic accuracy but most studies reported PPAs of >85% and NPAs of >98% (6). The FDA recommends pooled testing performances of >85% PPA when compared with the same test performed on individual samples (https://www.fda.gov/medical-devices/coronavirus-covid-19-and-medical-devices/pooled-sample-testing-and-screening-testing-covid-19). The developed systems fulfilled the FDA criteria regardless of the system or specimen type (≥90.5%; Table 4). The NPA values of the developed systems were very high (≥99.8%), indicating minimal contamination at the pooling stage. In Panther-PS and Biomek-PS testing, one and three pools that comprised negative samples tested positive in pooled testing but negative in individual testing (Table 4). These samples might indicate cross-contamination during the pooling stage. Their test results were reported correctly but a longer turnaround time was needed. Pooled testing using three systems did not produce invalid results. For quality control of pooled testing systems, it is important to monitor such conflicting or invalid testing results.

## Comparison with rapid antigen tests

Rapid antigen tests by lateral flow immunoassays can be used as point-of-care testing methods. They have been used for large-scale testing in clinics, hospitals, schools, testing centers, and nonmedical workplaces (23–26). The advantages of pooling molecular testing over rapid antigen tests include high sensitivity and the ability to detect pathogens for which rapid antigen tests are unavailable (e.g., at the early stage of the COVID-19 pandemic and future emerging pathogens). Our previous evaluation revealed that the LOD of the Fujirebio rapid antigen test was >1,000 fold greater than that of the three developed pooling systems (22). A retrospective study that analyzed >1 million subjects estimated the sensitivity of the Veritor rapid antigen test for asymptomatic individuals to be 38.5%–84.2% compared with that of the RT-PCR assay (27). Theoretically, the frequent use of rapid antigen tests may be effective for controlling the spread of disease in a mass population (28). However, for individuals who are tested, it is very important to know their status accurately for early treatment and to prevent transmission to their neighbors. We believe that both rapid antigen

tests and pooling molecular testing are screening test options and that a suitable approach can be selected depending on the following requirements and situations: disease prevalence, subject population, specimen types, testing capacity, turnaround time, analytical sensitivity, and reagent/equipment supply.

This study has several limitations. First, we used probit analysis to estimate the LODs because not all of the measurements around the LODs could be repeated 20 times for each pool/system size due to limitations in time, sample matrix, and funding. Second, pooled testing in the validation study was retrospectively performed using the samples tested with the reference assay. However, it is noted that most previous studies did not perform validation using clinical samples or performed using a limited number of samples. Third, in the validation study, different samples were used for each system, which made direct comparisons among the pooling systems impossible. Fourth, the validation study lacked validation by multiple investigators or in multiple locations, and samples were obtained from different clinical backgrounds. These factors might limit the generalizability of the findings to other laboratories and other patient populations. Fifth, we could not assess the clinical significance of false-negative samples by pooled testing due to a lack of clinical information. Sixth, we could not perform a comprehensive cost analysis including initial investment, maintenance, and operational costs over time. These costs could vary in different settings and regions.

In conclusion, we developed three different pooling molecular diagnostic systems that included automated media pooling devices, automated molecular detection platforms, and LISs. All of these systems were validated to perform reliable media pooled testing with automation and information management. Depending on the specimen types, needed testing capacity, required turnaround time, and availability of a fully automated molecular detection platform (Table 1), an appropriate system and pool size may be selected; Panther-PS or Biomek-PS may be suitable for large-scale screening, and geneLEAD-PS may be useful in clinical laboratories. The developed systems will increase the testing capacities of molecular diagnostic tests while saving costs and may contribute to infection control of COVID-19 and future emerging pathogens.

## MATERIALS AND METHODS

### Automated media pooling and molecular detection

In the pooling stage, an equal amount of liquid from each specimen in a pool was combined into a single tube or a plate well to give a final volume that was needed in the corresponding automated platform. Individual samples were retained for further testing if the pool tested positive. Customized pooling or testing protocols for Biomek 4000, JANUS G3, and Biomek i5, which were developed in this study, are available from the manufacturers or authors upon request. To avoid cross-contamination of the samples, the following measures were employed. An air gap (aspiration of air) was introduced after aspirating the liquid to prevent the liquid from dripping out of the disposable filter tip while moving to a new location for dispensing. Gloves and other personal protective equipment were used appropriately. Environmental decontamination, including decontamination of device surfaces, laboratory benches, and micropipettes, was performed using freshly prepared 0.5% sodium hypochlorite and 70% ethanol or ultraviolet irradiation. The pooling and testing stages were performed in separate negative pressure rooms. Waste was collected in disposable bags and autoclaved at a separate waste treatment facility. Air sampling was performed to ensure the absence of aerosol contamination.

For the molecular detection of SARS-CoV-2, geneLEAD VIII used an *in vitro* diagnostics assay LeaDEA VIASURE SARS-CoV-2 PCR Kit (Precision System Science), the Panther system used an *in vitro*-diagnostics assay Aptima SARS-CoV-2 (Hologic), and Biomek i5 used a laboratory developed test, the N2 assay (a Japanese reference assay approved for clinical diagnosis) (29). All assays followed the manufacturer's or developer's recommen-

dations, and the same interpretive criteria as individual testing were used for pooled testing (Table S3).

Testing results of pooled samples were automatically interpreted by LIS, which was developed in this study (Fig. 1): geneLEAD-PS used PSS LIS (Precision System Science), and the others used SimpPCR (Nippon Control System, Yokohama, Japan). If a pooled sample tested positive for SARS-CoV-2, each sample in that pool was then tested individually using the same detection platform to identify the positive sample(s). If the result was negative, all samples in that pool were considered negative without performing further tests. If the result was invalid, the pooled sample was retested.

## Samples

We included samples that were submitted to the laboratory at Kyoto University Hospital for SARS-CoV-2 RT−PCR testing between August 2020 and March 2021 for admission screening of COVID-19 or screening of contacts of COVID-19 clusters that occurred at healthcare facilities in Kyoto City, Japan. The samples that had an adequate remaining volume for the study were eligible and were prospectively stored at −80°C.

Viral transport media (UTM; Copan, Brescia, Italy) and flocked swabs (Copan) were used for the collection of anterior nasal and nasopharyngeal specimens. For Panther-PS, the Aptima Multitest Swab Specimen Collection Kit (cotton swab and proprietary inactivation buffer; Hologic) was used for the collection of nasal specimens. Saliva was collected by the saliva collection device Salisoft (Sarstedt, Nümbrecht, Germany) or straw and tube using the drooling technique followed by liquefaction with semialkaline protease (Kyokuto Pharmaceutical Industrial, Tokyo, Japan). Details of the samples used are shown in Table S4.

## Analytical sensitivity

We determined the LOD of each system using a minimum of 6 replicates of 2-fold or 10-fold serial dilutions of the heat-inactivated SARS-CoV-2 strain (ATCC VR-1986HK) of 100–10,000 genome copies/mL in a negative matrix (pooled saliva or nasal swab samples that tested negative by the reference assay). Lower or higher concentrations were assayed if the above concentrations could not define the LOD. We calculated the 95% LOD using probit analysis.

## Mock sample evaluation

A total of 200 saliva and 200 nasopharyngeal samples, including 10 positive saliva and 10 positive nasopharyngeal samples with Ct values 20–30 by the reference assay (Data set S1), were arbitrarily selected to assemble 10 positive and at least 10 negative pools for pool sizes of 4, 6, and 10. GeneLEAD-PS used a size of 8 instead of 10 because the maximum size of the pool supported by its pooling device was 8. These 400 samples were tested by each pooling system. Individual testing was also performed using each automated platform.

## Air sampling

To detect aerosols containing SARS-CoV-2 during media pooling, air sampling was conducted during mock sample evaluations. The air inside the pooling systems was vacuumed through a duct and collected into a cone containing 15 mL of sterile deionized water with 0.005% Triton X100 by a Coriolis Micro (Betrin Technologies, Montigny-le-Bretonneux, France) at a flow rate of 300 L/minute. Samples were tested by the reference assay.

## Validation study

Pooled testing was performed for consecutive stored samples of which test results of the reference assay were blinded to investigators but different samples were used

Microbiology Spectrum

for evaluation of each system due to limitation of sample volumes. The results were compared with the reference assay. We determined the size of the pool as the maximum number that fulfilled the following criteria: LOD ≤2,500 copies/mL, 100% concordance with the reference assay in the mock sample evaluation, and estimated negative pool fraction of >90%. To calculate the last parameter, the RT-PCR test positivity in the validation study was estimated as 1.4%, which was the test RT-PCR test positivity at the study laboratory from March to July 2020. Under this positivity and the assumption of all positive pools with only one positive sample, it was calculated that the needed pool size was <8.

## Cost and testing capacity

We calculated the costs for consumables and reagents for individual and pooled testing to estimate the cost reduction rates by pooled testing. Testing capacities per batch and following batches were estimated using the turnaround time.

## Reference RT-PCR assay

Individual respiratory samples were transferred to a 96-well deep-well plate using ASSIST PLUS (INTEGRA Biosciences, Tokyo, Japan) or JANUS G3. Viral RNA was extracted from 200 µL of samples using the MagNA Pure 96 Instrument and the MagNA Pure 96 DNA and Viral NA Small Volume extraction kit (Roche, Basel, Switzerland) and eluted in a final volume of 50 µL. Real-time RT-PCR was performed using the Japanese reference N2 assay (29) with TaqPath 1-Step RT−qPCR Master Mix, CG (Thermo Fisher Scientific). The RT-PCRs were performed using a LightCycler 480 System II (Roche, Basel, Switzerland), and cycle threshold (Ct) values were determined by the second derivative maximum method. The results were interpreted using the criteria in Table S3.

## Statistical analysis

The agreement between pooled or individual testing and the reference assay was assessed by Cohen's kappa concordance coefficient. The PPA and NPA of different samples were compared using Fisher's exact test. Associations between the Ct values of pooling and individual testing were analyzed using linear regression. A $P$ value < 0.05 was considered to indicate statistical significance. All statistical analyses were performed using SAS Studio 3.8 (SAS Institute Inc., Cary, NC). Visualization of the Ct values was conducted using R (https://cran.r-project.org) and ggplot2 (https://ggplot2.tidy-verse.org).

## ACKNOWLEDGMENTS

We thank Yosuke Kumano, Eiki Kure, Shoichi Nakai, Mizuki Mori, Kazuki Kitamura (Department of Clinical Laboratory Medicine, Kyoto University Graduate School of Medicine), Ryoji Karinaga, Kazumi Sawakami, Sayaka Kimura, Hideo Ikeda, Hideji Tajima (Precision System Science), Tomofumi Nakazaki (Beckman Coulter), Takuma Iwamatsu (Hologic), Kiyohide Iwashita (INTEGRA Biosciences), and Kanako Kitamori (Nippon Control System) for their technical assistance.

This research was supported by the Japan Agency for Medical Research and Development (AMED) under Grant Number JP20he1422001, joint research funds from Precision System Science, Beckman Coulter, Integra Biosciences, and Nippon Control System, and the COVID-19 Private Fund (to the Shinya Yamanaka laboratory, CiRA, Kyoto University). The funding organizations had no role in the study design, data collection, data analysis, data interpretation, or writing of the report.

Y.M.: conceptualization, data curation, formal analysis, funding acquisition, investigation, methodology, project administration, resources, supervision, validation, visualization, writing—original draft, writing—review & editing T.N.: investigation, writing—review & editing K.S.: investigation, writing—review & editing M.Y.: investigation, data

curation, resources, writing—review & editing M.N.: funding acquisition, resources, supervision, writing—review & editing.

## AUTHOR AFFILIATION

[1]Department of Clinical Laboratory Medicine, Kyoto University Graduate School of Medicine, Kyoto, Japan

## AUTHOR ORCIDs

Yasufumi Matsumura ⓘ http://orcid.org/0000-0001-8595-8944

## FUNDING

| Funder | Grant(s) | Author(s) |
|---|---|---|
| Japan Agency for Medical Research and Development (AMED) | JP20he1422001 | Yasufumi Matsumura |
| | | Masaki Yamamoto |
| | | Miki Nagao |
| Precision System Science | | Yasufumi Matsumura |
| | | Miki Nagao |
| Beckman Coulter Foundation | | Yasufumi Matsumura |
| Integra Biosciences | | Yasufumi Matsumura |
| Nippon Control System | | Yasufumi Matsumura |
| COVID-19 Private Fund (to the Shinya Yamanaka laboratory, CiRA, Kyoto University) | | Yasufumi Matsumura |
| | | Masaki Yamamoto |
| | | Miki Nagao |

## AUTHOR CONTRIBUTIONS

Yasufumi Matsumura, Conceptualization, Data curation, Formal analysis, Funding acquisition, Investigation, Methodology, Project administration, Resources, Supervision, Validation, Visualization, Writing – original draft, Writing – review and editing | Taro Noguchi, Investigation, Writing – review and editing | Koh Shinohara, Investigation, Writing – review and editing | Masaki Yamamoto, Data curation, Investigation, Resources, Writing – review and editing | Miki Nagao, Funding acquisition, Resources, Supervision, Writing – review and editing

## ETHICS APPROVAL

This study was performed in line with the principles of the Declaration of Helsinki. The Ethics Committee of Kyoto University Graduate School and the Faculty of Medicine approved this study (R2379) and waived the need to obtain informed consent from each patient.

## ADDITIONAL FILES

The following material is available online.

### Supplemental Material

**Tables S1 to S4, Figure S1 (Spectrum03684-23-s0001.docx).** Supplemental tables and figure.

### Open Peer Review

**PEER REVIEW HISTORY (review-history.pdf).** An accounting of the reviewer comments and feedback.

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
