## [Reviewer comments · Microbiology Spectrum]

Microbiology Spectrum

Development and evaluation of three automated media pooling and molecular diagnostic systems for the detection of SARS-CoV-2

Yasufumi Matsumura, Taro Noguchi, Koh Shinohara, Masaki Yamamoto, and Miki Nagao

Corresponding Author(s): Yasufumi Matsumura, Kyoto Daigaku

Review Timeline:

Submission Date:	October 19, 2023
Editorial Decision:	December 7, 2023
Revision Received:	December 21, 2023
Accepted:	December 22, 2023

Editor: Paul Luethy

Reviewer(s): Disclosure of reviewer identity is with reference to reviewer comments included in decision letter(s). The following individuals involved in review of your submission have agreed to reveal their identity: Xianding Deng (Reviewer #2)

Transaction Report:

DOI: <https://doi.org/10.1128/spectrum.03684-23>

Re: Spectrum03684-23 (Development and evaluation of three automated media pooling and molecular diagnostic systems for the detection of SARS-CoV-2)

Dear Dr. Yasufumi Matsumura:

Thank you for the privilege of reviewing your work. Below you will find my comments, instructions from the Spectrum editorial office, and the reviewer comments. The reviewers felt that this manuscript was backed by solid data, but had a few questions or revisions that they thought would solidify the manuscript.

Revision Guidelines

Sincerely,
Paul Luethy
Editor
Microbiology Spectrum

Reviewer #1 (Comments for the Author):

Summary

In this manuscript the authors describe the results of developing and comparing three pooling and testing symptoms for the screening of clinical samples for SARS-CoV-2. Their goal was to determine whether these systems could reliably detect SARS-CoV-2 so that time and money could be saved during situations like the pandemic where extremely large numbers of samples

need to be tested. Their study demonstrated that all three platforms are reliable for screening pooled clinical samples. Overall, the manuscript is well written, and the results of the study are clear. I only have some minor comments.

Minor Comments

1. In both the Abstract (line 40) and the Results (line 125) it says that the PPA was {greater than or equal to}95% and the NPA was {greater than or equal to}99.9% for all systems. However, in Table 4 it shows that the geneLEAD-PS system with nasopharyngeal swabs had a PPA of 90.5% and the same system with the nasal swabs had a NPA of 99.8%. So these statements and the table are not in agreement. The statements should be updated to reflect the data in the table or you should clarify whether you are only referring to certain samples types.
2. Similarly in the Discussion on line 225 it I believe it is trying to say the PPA was {greater than or equal to}92.9% based on Table 4. You should clarify if that is the PPA that you are referring to. And again shouldn't that be {greater than or equal to}90.5%?
3. In Table 2 it says that the limit of detection for the geneLEAD-PS system with individual testing was 62 copies/mL. However, in Table S1 it says that 100% of the replicates were positive at 100 copies/mL and only 17% positive at 50 copies/mL. How was the limit of detection of 62 copies/mL determined here? I thought probit required two concentrations that were neither 100% positive nor 0% positive.
4. In supplemental Table S1 there appears to be a mistake. For Panther-PS (saliva) with the individual testing and with a pool size of 10 some of the cells in the table have the viral copies/mL listed in the cell with the positive rate and replicates positive/tested. Where this occurs is with individual testing at 500, 250, and 100 viral copies/ml and a pool size of 10 with 25,000 and 10,000 copies/ml. Since the viral copies/mL are listed at the top of the table and not listed in any of the other cells I imagine these are not supposed to be listed here.

Reviewer #2 (Comments for the Author):

In Introduction, authors should review common strategies of sample pooling, is 10-samples or 5-samples method commonly used, in what settings? Sample pooling was also used for large-scale community screening, specially in China for outbreak containment, for example screening millions of residents in ShenZhen. These should be reviewed as practical applications of pooling.

What are differences among three automated pooling systems, simply different brands? Is automated pooling just a liquid handler (for pipetting and mixing)?

Why cannot Biomek4000 be paired with Biomeki5 in figure 1 (same brand)? Is pooling equipment independent from RT-PCR? so any one of three pooling machines can be paired with PCR detection equipment.

What measures authors used to avoid cross-contamination among samples during testing?

Why are pool sizes different among 3 systems, e.g. 6 for geneLEAD-PS, but 4 for other two? What is the reasoning behind?

In the discussion section, are these systems mainly used for clinical diagnostics of COVID-19, or large-scale community screening?

There are also other large-scale screening methods such as rapid antigen POC test, why do authors prefer pooling PCR over this?

Review

Summary

In this manuscript the authors describe the results of developing and comparing three pooling and testing symptoms for the screening of clinical samples for SARS-CoV-2. Their goal was to determine whether these systems could reliably detect SARS-CoV-2 so that time and money could be saved during situations like the pandemic where extremely large numbers of samples need to be tested. Their study demonstrated that all three platforms are reliable for screening pooled clinical samples. Overall, the manuscript is well written, and the results of the study are clear. I only have some minor comments.

Minor Comments

1. In both the Abstract (line 40) and the Results (line 125) it says that the PPA was $\geq 95\%$ and the NPA was $\geq 99.9\%$ for all systems. However, in Table 4 it shows that the geneLEAD-PS system with nasopharyngeal swabs had a PPA of 90.5% and the same system with the nasal swabs had a NPA of 99.8%. So these statements and the table are not in agreement. The statements should be updated to reflect the data in the table or you should clarify whether you are only referring to certain samples types.
2. Similarly in the Discussion on line 225 it I believe it is trying to say the PPA was $\geq 92.9\%$ based on Table 4. You should clarify if that is the PPA that you are referring to. And again shouldn't that be $\geq 90.5\%$?
3. In Table 2 it says that the limit of detection for the geneLEAD-PS system with individual testing was 62 copies/mL. However, in Table S1 it says that 100% of the replicates were positive at 100 copies/mL and only 17% positive at 50 copies/mL. How was the limit of detection of 62 copies/mL determined here? I thought probit required two concentrations that were neither 100% positive nor 0% positive.
4. In supplemental Table S1 there appears to be a mistake. For Panther-PS (saliva) with the individual testing and with a pool size of 10 some of the cells in the table have the viral copies/mL listed in the cell with the positive rate and replicates positive/tested. Where this occurs is with individual testing at 500, 250, and 100 viral copies/ml and a pool size of 10 with 25,000 and 10,000 copies/ml. Since the viral copies/mL are listed at the top of the table and not listed in any of the other cells I imagine these are not supposed to be listed here.

**Replies to the comments raised by the Editor**

Thank you very much for your review of our manuscript. We are grateful to the reviewers for
suggesting important modifications for the manuscript. We have carefully taken them into
consideration. Our point-by-point responses to the reviewers' suggestions are provided below.

**Replies to the comments raised by Reviewer #1**

Summary

In this manuscript the authors describe the results of developing and comparing three pooling and
testing symptoms for the screening of clinical samples for SARS-CoV-2. Their goal was to
determine whether these systems could reliably detect SARS-CoV-2 so that time and money could
be saved during situations like the pandemic where extremely large numbers of samples need to be
tested. Their study demonstrated that all three platforms are reliable for screening pooled clinical
samples. Overall, the manuscript is well written, and the results of the study are clear. I only have
some minor comments.

**Response.** Thank you very much for your review of the manuscript and for providing important
comments to improve our manuscript.

Minor Comments

1. In both the Abstract (line 40) and the Results (line 125) it says that the PPA was {greater than or
equal to}95% and the NPA was {greater than or equal to}99.9% for all systems. However, in Table
4 it shows that the geneLEAD-PS system with nasopharyngeal swabs had a PPA of 90.5% and the
same system with the nasal swabs had a NPA of 99.8%. So these statements and the table are not in
agreement. The statements should be updated to reflect the data in the table or you should clarify
whether you are only referring to certain samples types.

Response. In the original manuscript, the Abstract was written based on the overall respiratory
sample data, regardless of specimen type. In this setting, the overall PPAs/NPAs were described for
each system, which corresponded to the data described in the “All” specimen rows. Following your
comment, we modified the text to show the minimum PPA/NPA values and clarified that we referred
to any specimen type.

“All 3 systems showed high positive percent agreement values of $\geq 90.5\%$ and negative percent
agreement values of $\geq 99.98\%$ for any specimen type.” (In the revised, marked-up manuscript, line
39)

2. Similarly in the Discussion on line 225 it I believe it is trying to say the PPA was {greater than or
equal to}92.9% based on Table 4. You should clarify if that is the PPA that you are referring to. And
again shouldn't that be {greater than or equal to}90.5%?

Response. Thank you for pointing out this error. The PPA value should be 90.5%. In addition,

“pooling device” and “Table S3” are not needed here. We corrected the sentence as follows.

“The developed systems fulfilled the FDA criteria regardless of the system, ~~pooling device, and or~~
specimen type ($\geq 90.5\%$; Table 4).” (line 230)

3. In Table 2 it says that the limit of detection for the geneLEAD-PS system with individual testing
was 62 copies/mL. However, in Table S1 it says that 100% of the replicates were positive at 100
copies/mL and only 17% positive at 50 copies/mL. How was the limit of detection of 62 copies/mL
determined here? I thought probit required two concentrations that were neither 100% positive nor
0% positive.

Response. We agree with your points. The probit analysis requires a minimum of two concentrations
of nonzero or non-one hit rates for calculation. We added experiments for 75 copies/mL. Of the 10
replicates, 6 were determined to be positive (60%). The LOD was recalculated as 94 copies/mL
(95% confidence interval, 81–131). We thoroughly checked the LOD data (Table S1) and found that
the data for Panther-PS (saliva) with a pool size of 4 had only one nonzero or non-one hit rate. After
careful inspection, we found that the hit rates for 5,000 and 2,500 were incorrectly described. These
values were corrected to 83% (5/6) and 83% (5/6), respectively. Following this modification, the
LOD was recalculated as 5,552 copies/mL (95% confidence interval, 3,316–22,986).

4. In supplemental Table S1 there appears to be a mistake. For Panther-PS (saliva) with the
individual testing and with a pool size of 10 some of the cells in the table have the viral copies/mL
listed in the cell with the positive rate and replicates positive/tested. Where this occurs is with
individual testing at 500, 250, and 100 viral copies/ml and a pool size of 10 with 25,000 and 10,000
copies/ml. Since the viral copies/mL are listed at the top of the table and not listed in any of the
other cells I imagine these are not supposed to be listed here.

**Response.** Thank you for carefully reviewing the tables. We agree with your comments; these
additional concentrations should not be used. We corrected the table.

**Replies to the comments raised by Reviewer #2**

Thank you very much for your review of the manuscript and for providing valuable comments to
improve our manuscript.

In Introduction, authors should review common strategies of sample pooling, is 10-samples or
5-samples method commonly used, in what settings? Sample pooling was also used for large-scale
community screening, specially in China for outbreak containment, for example screening millions
of residents in ShenZhen. These should be reviewed as practical applications of pooling.

**Response.** Information on large-scale studies, including China’s city-wide community screening
involving >1 million people, was added to explain practical applications. Various pool sizes of 3, 5,
8, 10, and 24 have been used depending on the situation.

“However, it is noted that the majority of these studies involved only a mock sample evaluation.

~~Implementation of large-scale pooled testing, such as laboratory testing, has also been performed,~~
~~but the diagnostic performance has not yet been investigated (7-9).~~ Large-scale pooled testing

(approximately 40,000–250,000 samples) has already been implemented in clinical laboratories or

screening programs for schools or universities using pool sizes of 5, 8, 10, or 24, depending on

expected test positivity (7-9). China utilized pooled testing with pool sizes of 3, 5, or 10 for

community screening that targeted >1 million residents in moderate-risk areas, while people in

high-risk areas were tested individually to control local outbreaks (10, 11). However, in these

large-scale reports, the diagnostic performance of these methods was not investigated due to a lack

of comparisons with individual tests.” (line 81)

What are differences among three automated pooling systems, simply different brands? Is

automated pooling just a liquid handler (for pipetting and mixing)?

**Response.** We assume that you referred to the automated pooling devices (not the entire pooling

systems). The differences among the 3 pooling devices included acceptable sample formats (input
tubes), sample identification methods (barcode scanners), ranges of pool sizes, and pooling times.
Each automated pooling device included a liquid handler and a barcode reader. These data are
summarized in Table 1. To clarify the components of the pooling devices, we added the following
description to the legend of Figure 1.

“Each pooling device included a liquid handler and a barcode reader.”

Why cannot Biomek4000 be paired with Biomeki5 in figure 1 (same brand)? Is pooling equipment
independent from RT-PCR? so any one of three pooling machines can be paired with PCR detection
equipment.

**Response.** The 3 developed systems use different devices between the pooling and testing stages
(Figure 1), and the devices work independently. However, the combinations of devices for the
pooling and testing stages are currently limited to the pairs shown in Figure 1. Each pooling device
can handle only a specific type of input tube or output tube (or plate) due to the absence of an
appropriate tube rack or control program and lack of development/validation. For example, the
Aptima tube used in the Panther testing device is not compatible with the output tube of the Pooling
Mini Station or JANUS G3 pooling device. Theoretically, it is possible to configure JANUS G3 to
accept Aptima tubes, but the development and validation processes require considerable costs, labor,
and time. The input tube for the geneLEAD VIII testing system is not compatible with the output
tube of the Biomek 4000 or JANUS G3 system. It is possible to pair Biomek 4000 and Biomek i5,
but this combination requires another liquid handler with a barcode scanner (such as JANUS G3) to
inactivate samples before the pooling stage with Biome 4000, resulting in extra costs, labor and
time.

What measures authors used to avoid cross-contamination among samples during testing?

**Response.** To avoid cross-contamination of samples in the pooling stage, we employed liquid
handlers with disposable filter tips. An air gap (aspiration of air) was introduced after aspirating the
liquid to prevent the liquid from dripping out of the tip while moving to a new location for
dispensing. Gloves and other personal protective equipment were used appropriately. Environmental
decontamination, including device surfaces, laboratory benches, and micropipettes, was performed
using freshly prepared 0.5% sodium hypochlorite and 70% ethanol or ultraviolet irradiation. The
pooling and testing stages were performed in separate negative pressure rooms. Waste was collected
in disposable bags and autoclaved at a separate waste treatment facility. Air sampling was
performed to ensure the absence of aerosol contamination. This information was added to the
Methods section, and the results of the air sampling were added to the Results section. The methods
for air sampling were already described in the previous manuscript.

**“Air sampling**

**During the pooling procedures of each pooling device, 5 air samples were serially obtained. All the**
**samples were negative according to the reference assay.” (line 126)**

**“To avoid cross-contamination of the samples, the following measures were employed. An air gap**
**(aspiration of air) was introduced after aspirating the liquid to prevent the liquid from dripping out**
**of the disposable filter tip while moving to a new location for dispensing. Gloves and other personal**
**protective equipment were used appropriately. Environmental decontamination, including**
**decontamination of device surfaces, laboratory benches, and micropipettes, was performed using**
**freshly prepared 0.5% sodium hypochlorite and 70% ethanol or ultraviolet irradiation. The pooling**
**and testing stages were performed in separate negative pressure rooms. Waste was collected in**
**disposable bags and autoclaved at a separate waste treatment facility. Air sampling was performed**
**to ensure the absence of aerosol contamination.” (line 287)**

Why are pool sizes different among 3 systems, e.g. 6 for geneLEAD-PS, but 4 for other two? What
is the reasoning behind?

**Response.** To ensure the analytical sensitivity of each system, the pool sizes used in the validation
study were determined as the maximum number that fulfilled the following criteria: LOD \leq 2,500
copies/mL, 100% concordance with the reference assay in the mock sample evaluation, and an
estimated negative-pool fraction (a fraction of samples that were determined to be negative by only
testing pooled samples) $>$ 90%. To calculate the last parameter, the RT-PCR test positivity in the
validation study was estimated to be 1.4%, which was the test RT-PCR test positivity at the study
laboratory from March to July 2020. Under this assumption of positivity and the assumption of all
positive pools with only one positive sample, the needed sample size was calculated to be $<$ 8. This
process is described in the Methods section. The LOD and mock sample evaluation data are
described in the Results section (lines 116-121 with Tables 2 and 3) and were evaluated in the
Discussion section (lines 217-224).

In the discussion section, are these systems mainly used for clinical diagnostics of COVID-19, or
large-scale community screening?

**Response.** Depending on the specimen type, needed testing capacity, required turnaround time, and
availability of a fully automated molecular detection platform (Table 1), the appropriate system (and
pool size) varies. In general, high testing capacity is required for large-scale community screening,
while shorter turnaround times and high detection sensitivities are required in clinical laboratories.
Under these simplified assumptions, the Panther-PS or Biomek-PS is good for community screening,
and the geneLEAD-PS is good for clinical diagnostics. These preferences were added to the
Discussion section. In addition, “turn-around time” was changed to “turnaround time” to maintain
wording consistency throughout the manuscript.

“Depending on the specimen types, needed testing capacity, **required turnaround time**, and
availability of a fully automated molecular detection platform (Table 1), an appropriate system **and**
**pool size** may be selected-; **Panther-PS or Biomek-PS may be suitable for large-scale screening, and**
**geneLEAD-PS may be useful in clinical laboratories.”** (line 273)

There are also other large-scale screening methods such as rapid antigen POC test, why do authors
prefer pooling PCR over this?

**Response.** Rapid antigen tests have been employed in several countries as large-scale screening
tests (Parikh et al. Front Public Health. 2022; Peto et al. EClinicalMedicine 2021; Ludwick et al.
Glob Implement Res Appl. 2023). We believe that both our molecular systems and rapid antigen
POC tests (lateral flow assays) are screening test options. To combat pandemics, multiple options
must be available, and suitable approaches must be selected according to the relevant requirements
and situations (e.g., disease prevalence, subject population, specimen types, testing capacity,
turnaround time, analytical sensitivity, and reagent/equipment supply). The advantages of molecular
approaches (our systems) include high test sensitivity and the ability to detect pathogens for which
rapid antigen tests are unavailable (e.g., SARS-CoV-2 detection at the early stage of the COVID-19
pandemic and future emerging pathogens). In particular, the molecular detection approach is far
superior in terms of detection sensitivity (approximately 1,000 times). The LODs of the three
developed systems were <2,500 copies/mL, and that of the Fujirebio rapid antigen test was
2,500,000 copies/mL according to our evaluation (Matsumura et al. J Clin Virol Plus 2023). In our
previous study, the Fujirebio rapid antigen test detected 45% of the reference-assay-positive samples,
missing 55% of infected personnel. A retrospective study that analyzed >1 million subjects
estimated that the sensitivity of the Veritor rapid antigen test for asymptomatic individuals was
38.5–84.2% when compared with that of the RT–PCR assay (Parikh et al. Front Public Health.
2022). Theoretically, the frequent use of rapid antigen tests may be effective for controlling the

spread of disease in communities (Larremore et al. Sci Adv. 2021). However, for individuals who
are tested, it is very important to know their status accurately for early treatment and to prevent
transmission to their neighbors. In conclusion, the molecular approach has merits over the rapid
antigen test depending on the situation (and vice versa), and we would like to contribute to the
molecular approach by evaluating the 3 study systems. This information was summarized and added
as follows.

**“Comparison with rapid antigen tests**

**Rapid antigen tests by lateral flow immunoassays can be used as point-of-care testing methods.**
**They have been used for large-scale testing in clinics, hospitals, schools, testing centers, and**
**nonmedical workplaces (23-26). The advantages of pooling molecular testing over rapid antigen**
**tests include high sensitivity and the ability to detect pathogens for which rapid antigen tests are**
**unavailable (e.g., at the early stage of the COVID-19 pandemic and future emerging pathogens).**
**Our previous evaluation revealed that the LOD of the Fujirebio rapid antigen test was >1,000-fold**
**greater than that of the three developed pooling systems (22). A retrospective study that analyzed >1**
**million subjects estimated the sensitivity of the Veritor rapid antigen test for asymptomatic**
**individuals to be 38.5–84.2% compared with that of the RT–PCR assay (27). Theoretically, the**
**frequent use of rapid antigen tests may be effective for controlling the spread of disease in a mass**
**population (28). However, for individuals who are tested, it is very important to know their status**
**accurately for early treatment and to prevent transmission to their neighbors. We believe that both**
**rapid antigen tests and pooling molecular testing are screening test options and that a suitable**
**approach can be selected depending on the following requirements and situations: disease**
**prevalence, subject population, specimen types, testing capacity, turnaround time, analytical**
**sensitivity, and reagent/equipment supply.” (line 239)**

Re: Spectrum03684-23R1 (Development and evaluation of three automated media pooling and molecular diagnostic systems for the detection of SARS-CoV-2)

Dear Dr. Yasufumi Matsumura:

Thank you for making modifications to your manuscript following review. Your manuscript has been accepted, and I am forwarding it to the ASM production staff for publication. Your paper will first be checked to make sure all elements meet the technical requirements. ASM staff will contact you if anything needs to be revised before copyediting and production can begin. Otherwise, you will be notified when your proofs are ready to be viewed.

Sincerely,
Paul Luethy
Editor
Microbiology Spectrum